# Deep Learning Predicts Postoperative Mobility, Activities of Daily Living, and Discharge Destination in Older Adults from Sensor Data

**DOI:** 10.3390/s25165021

**Published:** 2025-08-13

**Authors:** Thomas Derya Kocar, Simone Brefka, Christoph Leinert, Utz Lovis Rieger, Hans Kestler, Dhayana Dallmeier, Jochen Klenk, Michael Denkinger

**Affiliations:** 1Institute for Geriatric Research at AGAPLESION Bethesda Ulm, Ulm University Medical Center, 89081 Ulm, Germany; simone.brefka@agaplesion.de (S.B.); christoph.leinert@agaplesion.de (C.L.); dhayana.dallmeier@agaplesion.de (D.D.); michael.denkinger@agaplesion.de (M.D.); 2Geriatric Center Ulm, Ulm 89073, Germany; 3Institute of Medical Systems Biology, Ulm University, 89081 Ulm, Germany; utz.rieger@uni-ulm.de (U.L.R.); hans.kestler@uni-ulm.de (H.K.); 4Department of Epidemiology, Boston University School of Public Health, Boston, MA 02118, USA; 5Institute of Epidemiology and Medical Biometry, Ulm University, 89081 Ulm, Germany; jochen.klenk@uni-ulm.de; 6Clinic for Geriatric Rehabilitation, Robert-Bosch Hospital, 70376 Stuttgart, Germany

**Keywords:** deep learning, machine learning, geriatrics, inertial measurement unit, IMU, sensors, postoperative recovery, functional assessment, discharge planning, automation

## Abstract

**Highlights:**

**What are the main findings?**
Deep learning models applied to lumbar IMU data predict postoperative mobility (CHARMI) and activities of daily living (Barthel Index) in older surgical patients, with R^2^ values of 0.65 and 0.70, respectively.Recommended discharge destinations were predicted with 82% accuracy using lumbar IMU data and deep learning.

**What is the implication of the main finding?**
IMU-based assessment and deep learning offer the potential for automated, objective, and continuous monitoring of functional recovery.

**Abstract:**

The growing proportion of older adults in the population necessitates improved methods for assessing functional recovery. Objective, continuous monitoring using wearable sensors offers a promising alternative to traditional, often subjective assessments. This study aimed to investigate the utility of inertial measurement unit (IMU)-based data, combined with deep learning, to predict postoperative mobility, activities of daily living, and discharge destination in older adults following surgery. Data from the SURGE-Ahead project was analyzed, involving 39 patients (mean age 79.05 years) wearing lumbar IMU sensors for up to five postoperative days. Deep learning models (TabPFN) were applied and validated using leave-one-out cross-validation to predict the Charité Mobility Index (CHARMI), the Barthel Index, and discharge destination. The TabPFN model achieved R^2^ values of 0.65 and 0.70 for predicting CHARMI and Barthel Index scores, respectively, with moderate to strong agreement with human assessments (weighted kappa ≥ 0.80). Discharge destination was predicted with an accuracy of 82%. The z-channel IMU data and parameters related to walking bouts were most predictive of outcomes. IMU-based data, combined with deep learning, demonstrates potential for automated functional assessment and discharge decision support in older adults following surgery.

## 1. Introduction

The growing proportion of older adults in the population presents significant challenges to healthcare systems worldwide, characterized by a rising incidence of multimorbidity, functional decline, and subsequent healthcare needs [1,2]. Following surgery, particularly in the geriatric population, accurate assessment of recovery trajectories is crucial for optimizing patient management, resource allocation, and ultimately, achieving successful discharge [3,4]. Traditionally, assessing these outcomes relies on clinical observation and periodic manual assessments, which can be subjective, time-consuming, and fail to capture the oscillations of real-world functional performance.

Recent advancements in wearable sensor technology, specifically inertial measurement units (IMUs), offer a promising avenue for objective and continuous monitoring of patient mobility and activity levels [5,6,7]. IMUs provide a plethora of data regarding movement patterns, enabling the quantification of gait parameters, activity engagement, and functional performance with increasing precision [8,9]. Furthermore, the potential to leverage this data for predictive modelling is gaining momentum, with applications ranging from fall risk assessment to potentially predicting rehabilitation outcomes [10,11,12].

The application of machine learning and deep learning techniques to IMU-derived data has demonstrated remarkable success in various healthcare domains. Recent studies have shown the capability to accurately predict gait parameters and assess frailty [11,13]. However, the translation to the specific context of orthogeriatric patients requires careful consideration. In addition, leveraging IMU-derived functional parameters to inform clinical decision-making requires further investigation.

This study aims to investigate the utility of IMU-based data, combined with artificial intelligence (AI), to predict key outcomes following orthogeriatric surgery, including mobility, activities of daily living (ADL), and discharge destination. By leveraging the continuous, objective data provided by IMUs and the predictive power of deep learning, we seek to provide a proof-of-concept that explores the potential for automated clinical decision-making, optimizing rehabilitation strategies, and ultimately improving patient outcomes in this vulnerable population.

## 2. Materials and Methods

### 2.1. Study Population

This study constitutes a secondary analysis of data collected during the Supporting Surgery with Geriatric Co-Management and AI Observational and AI Development (SURGE-Ahead) study [14], conducted at three surgical departments of Ulm University Medical Center between February 2023 and March 2024. Included patients were aged 70 years and older, undergoing surgery, and had an Identification of Seniors at Risk (ISAR) score of 2 or higher. Baseline characteristics were assessed using several standardized tools. Frailty and multimorbidity were evaluated using the Clinical Frailty Scale (CFS), American Society of Anesthesiologists (ASA) score, Identification of Seniors at Risk (ISAR) score, and number of medications. Cognitive function was assessed with the Montreal Cognitive Assessment 5-min (MoCA), and dementia status was recorded. Nutritional status was assessed using the Nutritional Risk Screening (NRS) score. Psychological distress was assessed with the Patient Health Questionnaire-4 (PHQ-4). Functional status was measured using the Barthel Index [15], and mobility was further assessed using the Charité Mobility Index (CHARMI; [16]) and the New Mobility Score (NMS). Additional data collected included age, sex, type of surgery, emergency surgery status, body mass index (BMI), care level, and history of falls (for details, see study protocol [14]). Of 178 patients enrolled in the study, 169 completed the study protocol until discharge (n = 3 drop-outs; n = 6 deaths). A subset of 39 participants wore an IMU (Axivity AX6; axivity Ltd., Newcastle upon Tyne, UK) in the lumbar region for at least 24 h, with 22 concurrently completing a movement diary. Sensors were attached to the participants’ lumbar spine using an adhesive patch on postoperative day 1, with recording starting at 00:00 the following day and detached on postoperative day 5 or earlier if the patient was discharged. The X-axis of the sensor was oriented vertically, the Y-axis horizontally, and the Z-axis pointed towards the patient’s abdomen, as illustrated in the Appendix A. The sensor operated at a sampling rate of 100 Hz, with a resolution of 16 bits, and a battery lasting approximately 7 days. The Axivity AX6 contains a tri-axial accelerometer and gyroscope; however, in the current study, only the accelerometer data were analyzed. All participants underwent a comprehensive geriatric assessment. On postoperative day 3, assessments included the CHARMI and Barthel Index for mobility and ADL, respectively. In addition, discharge recommendations (back home, acute geriatric care unit, post-acute rehabilitation facility, or nursing home) were independently provided by two geriatricians, considering clinical assessment, patient preferences, and rehabilitation potential, and reconciled through joint review with 3-month follow-up data. For a STROBE chart, see the Appendix A.

### 2.2. IMU Feature Extraction

IMU sensor data was preprocessed using the scikit-digital-health 0.17.2 library [17] for Python 3.12.2 (Python Software Foundation). The DETACH algorithm [18] was used to detect IMU-wearing episodes in each participant, and data was sliced accordingly. For each IMU channel (x, y, z), a comprehensive set of features was extracted across the entirety of the wearing episode, using the features class from the scikit-digital-health library. Signal distribution was characterized by calculating the mean, standard deviation, skewness, kurtosis, range, interquartile range, range count percentage, and the ratio beyond r sigma. Time domain analysis included the mean cross rate, autocorrelation, signal entropy, jerk metric, root mean square value, and power. Frequency domain characteristics were assessed using dominant frequency, power spectral sum, power range sum, spectral flatness, and spectral arc length. Furthermore, we calculated the vector magnitude for the root mean square value, power, and power spectral sum. All basic signal features were extracted using the scikit-digital-health library’s default parameters.

Beyond basic signal features, we utilized the Sit-2-Stand algorithm [19] and the Mobilise-D pipeline [8,9], implemented within the scikit-digital-health 0.17.2 and mobgap 0.10.0 libraries [20] for Python (Python Software Foundation), respectively. The Sit-2-Stand algorithm analyzes transitions from a sitting to a standing position, yielding features including episode count, duration, maximum and minimum acceleration, spectral arc length, and displacement. The Mobilise-D pipeline identifies walking bouts and calculates their duration, cadence, stride duration, and walking speed, along with their average, variance, and maximum values (also see Appendix A). For more information on the Mobilise-D project, please refer to the project page (https://mobilise-d.eu/, accessed on 28 June 2025) and associated publications [8,9].

### 2.3. Gait Detection

Gait detection performance of the Mobilise-D pipeline was assessed using movement diaries from 22 participants. Documented ambulation events defined ‘true positive’ walking bouts; instances of pipeline detection outside these events were ‘false positives,’ and missed events were ‘false negatives.’ Precision, recall, and F1 scores (a balanced measure that considers both false positives and false negatives, particularly useful with imbalanced datasets). To determine the relationship between detection performance and patient mobility, F1 scores were correlated with the number of detected walking bouts, CHARMI scores, and Barthel Index scores, by calculating Spearman’s Rho.

### 2.4. Deep Learning

This study followed the TRIPOD + AI reporting guidelines [21] (see Appendix A) to ensure transparency and reproducibility. To predict postoperative outcomes, we employed TabPFN [22], a prior-fitted transformer designed for small tabular datasets, requiring no preprocessing or hyperparameter tuning, and capable of handling missing data. As a generative foundation model for tabular data, it was trained across millions of synthetic datasets, enabling it to learn robust representations and perform well with limited and complex data often encountered in geriatric populations. TabPFN’s prior-fitting approach reduces the risk of overfitting and allows for meaningful analysis even with relatively small sample sizes, a significant advantage in the context of geriatric research where data acquisition can be challenging. This contrasts with traditional deep learning methods that require extensive retraining and large datasets to achieve reliable performance. The model was utilized to predict CHARMI and Barthel Index scores at postoperative day 3 (using regression) and to classify optimal discharge destination (a multiclass task). Prior to model application, a feature selection pipeline utilizing Spearman correlation (for regression) and the Kruskal–Wallis test (for classification) identified significant features (*p* < 0.05) from the initial set of 94 extracted features. Model performance was evaluated using leave-one-out cross-validation, with R^2^, mean error, and weighted kappa (an inter-rater metric) for regression, and accuracy and receiver operating characteristic area under the curve (ROC AUC) micro average for classification. Finally, 95% confidence intervals (CI) were estimated via bootstrapping with 1000 iterations.

## 3. Results

### 3.1. Study Population

The study cohort comprised 39 patients (mean age 79.05 years ±6.4, 43.6% male) undergoing surgery at Ulm University Medical Center. The majority (74.4%) had undergone trauma surgery. The average BMI was 26.52 kg/m^2^ (±4.69), and the mean number of medications taken was 8.87 (±3.5), highlighting the significant multimorbidity present in the population. All demographic and clinical characteristics are displayed in Table 1. IMU sensors were worn for an average of 3.10 days (±1.41) following surgery, starting on postoperative day 1. At postoperative day 3, the mean CHARMI score was 4.91 (±2.84) and the mean Barthel Index score 59.46 (±24.32). Discharge recommendations favored home discharge for 22 patients; 16 were recommended for acute geriatric care, one for rehabilitation, and none for a nursing home.

### 3.2. Gait Detection

The Mobilise-D pipeline [8,9] detected walking bouts with a precision of 0.51, a recall of 0.87, and an F1 score of 0.64 across all events. Notably, the F1 score showed a highly significant correlation with several mobility-related parameters, including the number of walking bouts, CHARMI scores, and Barthel Index scores (*p* < 0.001; see Figure 1).

### 3.3. Deep Learning

The feature selection revealed significant associations between IMU features and both CHARMI and Barthel Index scores, as well as with recommended discharge destinations. The strongest associations were observed in the IMU z-channel, which records acceleration during forward/backward motion and, importantly, gravitational pull when the patient is lying on their back (supine position). Furthermore, parameters related to walking bouts also demonstrated significant associations with all outcomes. Detailed results of the exploratory analysis are available in the Appendix A.

TabPFN demonstrated accurate prediction of both CHARMI and Barthel Index scores at postoperative day 3. For CHARMI scores, the model achieved an R^2^ of 0.65 (95% CI: 0.34–0.80), a mean error of 1.32 (95% CI: 1.01–1.66), and a weighted kappa of 0.80 (95% CI: 0.64–0.89). For the Barthel Index, the corresponding values were an R^2^ of 0.70 (95% CI: 0.53–0.82), a mean error of 10.87 (95% CI: 8.71–13.23), and a weighted kappa of 0.81 (95% CI: 0.69–0.88). A visual comparison of estimated and assessed scores is presented in Figure 2. TabPFN also yielded an accuracy of 0.82 (95% CI: 0.69–0.92) and an ROC AUC micro-average of 0.86 (95% CI: 0.74–0.96) for predicting the recommended discharge destination.

## 4. Discussion

This study presents a proof-of-concept for automated functional assessments and discharge decision support in older adults following surgery, utilizing IMU sensors and deep learning.

Good predictions of both mobility (CHARMI; R^2^ = 0.65) and ADL (Barthel Index; R^2^ = 0.70) were achieved using a single lumbar sensor. Furthermore, an inter-rater analysis demonstrated moderate to strong agreement (weighted kappa ≥ 0.80) between predicted and actual assessment scores [23], highlighting the potential for automated functional assessment in geriatric inpatients post-surgery, although it should be noted that functional assessments provide detailed clinical information beyond a composite score, informing individualized care planning. A single lumbar IMU sensor and deep learning also accurately predicted the recommended discharge destination in 82% of cases, representing a novel application of IMU data analysis. These results are comparable to those achieved by models relying primarily on electronic health record data [24] and are particularly encouraging given the challenges of optimal discharge planning in older adults and the frequent occurrence of undertreatment. Observational data from the SURGE-Ahead project indicated a standard-of-care accuracy of 0.73 compared to geriatric expert recommendations [14], suggesting a potential for improvement through the application of artificial intelligence.

Notably, the Z-channel data from the IMU sensor showed the most significant association with mobility, ADL scores, and the recommended discharge destination. In contrast, analyses of the X- and Y-axis did not reveal comparable associations with discharge destinations (see Appendix A). When correctly positioned on the lumbar region, the Z-channel exhibits heightened activity during periods of patient recumbency, potentially indicating immobility and a bed-ridden state. Conversely, data derived from the walking bout analysis—reflecting a state of mobility—were similarly predictive in our exploratory analyses (see Appendix A).

Analysis of movement diaries from 22 patients demonstrated reliable identification of walking bouts using the Mobilise-D pipeline, achieving an F1 score of 0.64. This is lower than the F1 score of 0.77 found by Kirk and colleagues (2024) [9], likely attributable to the higher proportion of immobilized patients within our study cohort: Correlation analyses indicated a reduction in walking bout detection F1 scores among patients exhibiting few detected walking bouts, limited mobility, and impaired self-care capabilities (see Figure 1). We also performed an additional analysis excluding algorithmic walking bout detection, utilizing only raw sensor data (see Appendix A). Notably, this approach yielded inferior predictive performance (mobility R^2^ 0.52, ADL R^2^ 0.39, discharge destination accuracy 0.79), suggesting that, while imperfect, the walking bout detection algorithm contributes valuable information to the overall model. Analysis of IMU wearing time revealed no significant correlation with discharge destination or outcome scores, suggesting that the algorithm’s predictive performance is not primarily driven by the length of sensor use, but rather by the kinematic data acquired during that period (see Appendix A).

During the observational and AI development study [14], approximately one-quarter (n = 46) agreed to wear an IMU sensor, others declined participation due to postoperative immobilization and pain that hindered sensor application. During the sensor application process, several participants (n = 7) expressed discomfort with lumbar sensor placement. While all continued their participation in the study, these seven patients opted to wear the IMU sensor on their right thigh instead. Data collected from the thigh-mounted sensors could not be included in this analysis, as the Mobilise-D algorithm is not designed to process data from alternate anatomical locations. Furthermore, the time required for sensor attachment and data retrieval (typically 10–20 min) represents a potential limitation and must be taken into account when evaluating the overall efficiency gains afforded by automated assessment and discharge decision support. Occasional instances of accidental sensor removal by patients, who mistook the device for a wound dressing, also presented a logistical challenge. Consequently, the SURGE-Ahead study group is now exploring the use of wrist-worn sensors in a follow-up study, an approach supported by emerging evidence. Recent research by Kluge and colleagues (2024) supports the validity of using wrist-worn sensors to detect gait sequence in a variety of patient populations [6], including those recovering from fractures. While performance was generally lower than with sensors placed at the lower back, the study demonstrated acceptable sensitivity (0.55–0.81) and specificity (0.95–0.98) compared to a multi-sensor reference system.

The ultimate goal of the research project is to develop a tool to augment clinical decision-making. Defining acceptable performance thresholds requires careful consideration of the prediction’s risk and the level of human oversight. Fully automated assessments, such as ADL evaluation, would necessitate high accuracy, approaching inter-rater reliability between clinicians (weighted kappa ≥ 0.9). However, even a model with moderate predictive power can be valuable if it aids clinician insights or outperforms current standards of care, as demonstrated by our discharge destination predictions. Future research should focus on refining accuracy, integrating explainability methods for human supervision, and validating performance across diverse clinical applications.

### Limitations

This study represents a proof-of-concept investigation and should be interpreted considering several limitations. First, the relatively small sample size (n = 39) limits the generalizability of our findings. Consequently, the reported R^2^ and AUC values should be interpreted with consideration of the wide confidence intervals, which accurately reflect the uncertainty associated with these estimates. Second, the limited availability of IMU data (approximately 20% of the cohort) introduces potential bias and further constrains the statistical power of our analysis. Third, a significant limitation is the substantial class imbalance observed in discharge destinations, with no patients recommended for nursing home care. This imbalance poses challenges for model training and requires cautious interpretation of accuracy scores, as a majority class prediction can yield deceptively high overall accuracy. While calibration techniques can mitigate some effects, accurate prediction of minority class cases remains a concern. Fourth, the reliance on patient-reported events, which are subject to bias and potential inaccuracies, is a potentially limiting factor; the absence of objective, autonomous event verification techniques, such as image-based analysis, may have impacted the accuracy of the results. Fifth, while weighted kappa values demonstrated moderate to strong agreement with human assessments, they were lower than the 0.90–0.96 range typically observed between different human Barthel Index assessors [25]. Sixth, the observational nature of data collection led to variability in sensor wearing time periods, introducing potential bias. This will be addressed in future studies through the implementation of a standardized protocol with fixed sensor wearing durations. Seventh, this analysis was largely exploratory, and while deep learning techniques demonstrated promising predictive capabilities, further research is required to identify the optimal modeling approach. Eighth, some patients may have been bed-ridden, resulting in no walking bout episodes and a corresponding lack of related feature extraction. However, the TabPFN model automatically handles missing data, and the absence of walking bout data itself provides information regarding patient mobility. Finally, explainability of the model’s predictions was not explored and remains an important area for future research. Providing clinicians with a satisfactory understanding of the model’s reasoning as well as possible reasons for inaccurate predictions will be crucial for establishing trust and facilitating clinical adoption. While attribution methods such as SHAP may identify important features [26], these often correspond to low-level sensor signals that lack inherent clinical meaning and are not readily interpretable by clinicians.

## 5. Conclusions

This study demonstrates the potential of leveraging IMU sensor data and deep learning techniques for objective, automated assessment of functional recovery and discharge planning in older adults following surgery. Our findings support further investigation of IMU-based technologies as a valuable tool to optimize rehabilitation strategies and improve patient outcomes in this vulnerable population, ultimately paving the way for more personalized and efficient geriatric care.

## Figures and Tables

**Figure 1 sensors-25-05021-f001:**
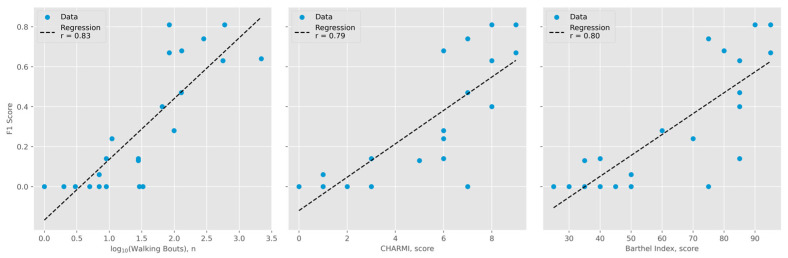
Gait detection performance compared to activity level. Shared Y-axis represents F1 scores. (**Left**): number of walking bouts detected by the Mobilise-D algorithm (logarithmic scale, base 10; r = 0.83). (**Middle**): Charité Mobility Index (CHARMI) scores (r = 0.79). (**Right**): Barthel Index scores (r = 0.80).

**Figure 2 sensors-25-05021-f002:**
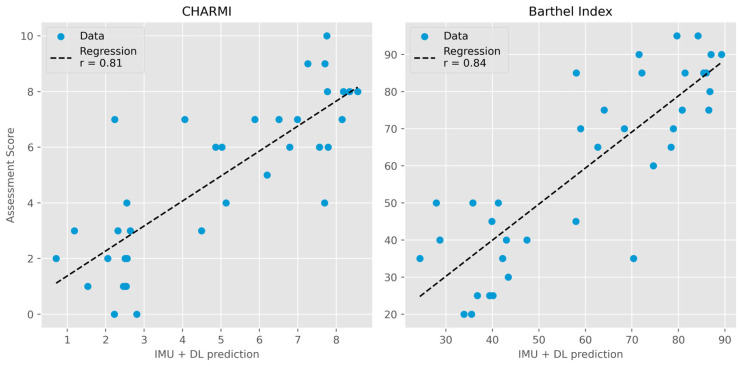
Deep learning prediction of assessment scores compared to actual scores. (**Left**): Charité Mobility Index (CHARMI; r = 0.81). (**Right**): Barthel Index (r = 0.84). IMU = inertial measurement unit. DL = deep learning (TabPFN).

**Table 1 sensors-25-05021-t001:** Study Population Characteristics. BMI = body mass index, NRS = nutritional risk screening, MoCA = Montreal Cognitive Assessment, PHQ4 = Patient Health Questionnaire-4, ISAR = identification of seniors at risk, CFS = Clinical Frailty Scale, ASA = American Society of Anesthesiologists score, CHARMI = Charité Mobility Index, NMS = New Mobility Score, pre-OP = preoperative, post-OP = postoperative.

	Mean (±sd)	n (%)
Age (years)	79.05 (±6.4)	
Sex (male)		17 (43.6%)
Trauma Surgery (Yes/No)		29 (74.4%)
General & Visceral Surgery (Yes/No)		3 (7.7%)
Urology (Yes/No)		7 (17.9%)
Emergency OP (Yes/No)		19 (48.7%)
BMI (kg/m^2^)	26.52 (±4.69)	
NRS (Yes/No)		13 (33.3%)
MoCA 5-min (score)	21.26 (±6.39)	
Dementia (Yes/No)		2 (5.1%)
PHQ4 (score)	2.67 (±2.45)	
ISAR (score)	2.97 (±1.01)	
CFS (score)	3.36 (±1.68)	
ASA (class)	1.84 (±0.49)	
Number of Medications (n)	8.87 (±3.5)	
Care Level (class)	0.51 (±0.91)	
Living Alone (Yes/No)		16 (41.0%)
Barthel Index pre-OP (score)	93.59 (±11.24)	
Barthel Index post-OP day 3 (score)	59.46 (±24.32)	
CHARMI pre-OP (score)	7.51 (±3.24)	
CHARMI post-OP day 3 (score)	4.91 (±2.84)	
NMS (score)	7.10 (±2.06)	
Falls (Yes/No)		23 (59.0%)
Sensor Worn (days)	3.10 (±1.41)	

## Data Availability

Data used in this specific analysis can be accessed on the project’s GitHub page: https://github.com/IfGF-UUlm/SA_Sensor-lumbar/, accessed on 28 June 2025. Raw IMU sensor data is not available on the project’s GitHub page due to file size limitations but is available upon request. Further information, data and results from the SURGE-Ahead observational and AI development study can be accessed on the project’s GitHub page: https://github.com/IfGF-UUlm/SA_OKIE/, accessed on 28 June 2025.

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
