# Peer review of "Deep Learning Predicts Postoperative Mobility, Activities of Daily Living, and Discharge Destination in Older Adults from Sensor Data"

_sensors, 2025, doi:10.3390/s25165021_

Round 1

Reviewer 1 Report

Comments and Suggestions for Authors

This manuscript reports on an interesting experiment to see how lumbar-worn IMU data can predict postoperative function and discharge destination.  The sample size was 39 patients who underwent various surgeries. The geriatric sample with a variety of surgeries and co-morbidities is highly appealing for tools that are to be eventually applied to clinical care. The main findings were that: 1) more than half of the variance in mobility (CHARMI) and activities of daily living (Barthel) could be predicted by IMU data; 2) discharge destination prediction was > 80%; and 3) it was the y-axis variables and variables related to walking bouts that were the most useful features. There are several issues below to address in revision.

Major

  1. A sample size of N = 39 is very small for a deep learning analysis. While leave-one-out cross validation is helpful, it does not negate the sample size concern. It was surprising to see that only around 20% of the patients in the study had IMU data. Please provide a bit more information about why such a small portion had the data. Was this because that is all that had usable data? Or other reasons? The small sample size shows proof of concept but decreases the value of the specific findings. While there are several places in the manuscript that modifiers such as “proof-of-concept” and “potential for” are carefully inserted, it is strongly recommended that the authors go through the manuscript carefully to increase the frequency of these disclaimers. In the Limitation section of the Discussion, please explicitly mention that consequence of many of these limitations is likely overestimations of R2 and AUC in the models.
  2. The walking bout detection performance was done on only 22 individuals and had relatively poor average F1 and precision values, despite their statistical significance. The inaccuracies in detection at the individual participant level can be seen clearly in Figure 1 scatterplots and Supplemental Table. Since the ultimate goal is more efficient clinical care, where decisions occur at the level of the individual (not the sample or population), it would be important for readers to know what R2 values are achieved without having to use the detection algorithms. This could be included as an additional supplement.
  3. Related to the point above, the authors are encouraged to expand the Discussion paragraph starting on line 197 to speculate about what statistics would make a model such as this acceptable for clinical utility at the individual patient level. How high would the R2s and AUCs need to be? Why?
  4. Please expand the discussion of generalizability of models with only N = 39. This is particularly important for the discharge destination model. As mentioned in the limitations, the data set contained only 3 of 4 possible discharge destinations and was heavily skewed (22/39) towards discharge home. Four categories by chance alone is 25%. If the model predicted home for everyone, then accuracy would already be at 56% in this sample.
  5. Somewhere in the Discussion, it is important to mention that there is utility of the actual functional assessments beyond the scores. These assessments often provide key information at the individual item level (e.g. this person needs assistance for toileting but not for stairs, standardized observation of how someone performs the activity to guide rehabilitation treatment) beyond the total score for care team. If the CHARM is only a 1 item scale by observation, it is going to be hard to justify a complex, digital solution that is easier. This information could be included in the Discussion paragraph starting on line 223, where the authors logically debate the pros and cons of the whole endeavor.

Minor

  1. In Table 1, this reviewer is highly appreciative of the descriptive statistics from many clinical scales and other aspects of care. There is no information about these scales in the Methods section. Providing more information about the scales would make the paper more accessible to those are not familiar with them or with clinical care. Ideally, the Methods would include a few short sentences about each one. If space does not permit that, then another option would be to add the appropriate references to the abbreviations at the top of Table 1.
  2. Consider shifting the focus in the text from highlighting statistical significance to highlighting the actual model statistics with their wide confidence intervals. Since a goal is to move forward to clinical care, then the relevance is not with the p value itself but with the variance explained and the range of error at the individual level.

Author Response

This manuscript reports on an interesting experiment to see how lumbar-worn IMU data can predict postoperative function and discharge destination. The sample size was 39 patients who underwent various surgeries. The geriatric sample with a variety of surgeries and co-morbidities is highly appealing for tools that are to be eventually applied to clinical care. The main findings were that: 1) more than half of the variance in mobility (CHARMI) and activities of daily living (Barthel) could be predicted by IMU data; 2) discharge destination prediction was > 80%; and 3) it was the y-axis variables and variables related to walking bouts that were the most useful features.

We thank the reviewer for appreciating the clinical potential of our work and its relevance to a complex geriatric population.

There are several issues below to address in revision.

Major

A sample size of N = 39 is very small for a deep learning analysis. While leave-one-out cross validation is helpful, it does not negate the sample size concern. It was surprising to see that only around 20% of the patients in the study had IMU data. Please provide a bit more information about why such a small portion had the data. Was this because that is all that had usable data? Or other reasons? The small sample size shows proof of concept but decreases the value of the specific findings. While there are several places in the manuscript that modifiers such as “proof-of-concept” and “potential for” are carefully inserted, it is strongly recommended that the authors go through the manuscript carefully to increase the frequency of these disclaimers. In the Limitation section of the Discussion, please explicitly mention that consequence of many of these limitations is likely overestimations of R2 and AUC in the models.

We agree with the reviewer's assessment regarding sample size limitations, particularly in the context of a deep learning analysis. However, we would like to highlight that we employed TabPFN, a prior-fitted transformer specifically designed for small tabular datasets. Unlike traditional deep learning models requiring extensive retraining, TabPFN leverages its pre-trained knowledge base, making it robust even with limited data and potentially capable of identifying patterns that might be missed by other approaches. Regarding the availability of IMU data in only approximately 20% of patients, we clarify that participation in IMU data collection was entirely optional. Many patients expressed concerns regarding sensor comfort, and while seven participants initially consented, they ultimately preferred alternative sensor placement on the right thigh. Unfortunately, data collected from the alternate location could not be integrated into this analysis due to the Mobilise-D Algorithm’s requirement for lumbar placement. We fully agree that these limitations, including the small sample size and restricted data availability, require careful consideration and will be explicitly addressed in the revised manuscript, with particular emphasis on the wide confidence intervals surrounding our model statistics, reflecting the inherent uncertainty in our findings.

We revised the manuscript, accordingly:

TabPFN’s prior-fitting approach reduces the risk of overfitting and allows for meaningful analysis even with relatively small sample sizes, a significant advantage in the context of geriatric research where data acquisition can be challenging. This contrasts with traditional deep learning methods that require extensive retraining and large datasets to achieve reliable performance.”

"During the observational and AI development study [14], approximately one-quarter (n=46) agreed to wear an IMU sensor, others declined participation due to postoperative immobilization and pain that hindered sensor application. During the sensor application process, several participants (n=7) expressed discomfort with lumbar sensor placement. While all continued their participation in the study, these seven patients opted to wear the IMU sensor on their right thigh instead. Data collected from the thigh-mounted sensors could not be included in this analysis, as the Mobilise-D algorithm is not designed to process data from alternate anatomical locations."

"First, the relatively small sample size (n=39) limits the generalizability of our findings. Consequently, the reported R² and AUC values should be interpreted with consideration of the wide confidence intervals, which accurately reflect the uncertainty associated with these estimates. Second, the limited availability of IMU data (approximately 20% of the cohort) introduces potential bias and further constrains the statistical power of our analysis."

In addition, supplementary figure S1 was revised to include this information.

The walking bout detection performance was done on only 22 individuals and had relatively poor average F1 and precision values, despite their statistical significance. The inaccuracies in detection at the individual participant level can be seen clearly in Figure 1 scatterplots and Supplemental Table. Since the ultimate goal is more efficient clinical care, where decisions occur at the level of the individual (not the sample or population), it would be important for readers to know what R2 values are achieved without having to use the detection algorithms. This could be included as an additional supplement.
Related to the point above, the authors are encouraged to expand the Discussion paragraph starting on line 197 to speculate about what statistics would make a model such as this acceptable for clinical utility at the individual patient level. How high would the R2s and AUCs need to be? Why?

We agree with the reviewer's concerns regarding the walking bout detection performance and the importance of individual-level accuracy for clinical utility. To address this, we have conducted an additional analysis utilizing only raw sensor data, bypassing the algorithmic walking bout detection step, and have included these results as supplementary table S4. We acknowledge the relatively modest F1 and precision values observed in the original analysis. Determining a definitive threshold for acceptable model performance in a clinical setting is complex, as it hinges on factors such as the presence of human supervision (“human-in-the-loop”) and the criticality of the prediction. We believe that even with inherent uncertainty, our model may offer clinical value, particularly in scenarios where the current standard of care yields poorer performance, as demonstrated by the accuracy of discharge destination predictions. Furthermore, if a human-in-the-loop approach is adopted, the implementation of robust explainability methods is crucial to enhance supervisory capabilities and mitigate the risk of erroneous decisions. For fully automated assessment of mobility in activities of daily living (ADL), achieving inter-rater agreement between machine and human assessors comparable to that observed between human assessors themselves is paramount. Literature suggests a weighted kappa of at least 0.9 for reliable ADL assessment, a target we believe may be attainable with larger datasets.

The manuscript was revised accordingly:

We also performed an additional analysis excluding algorithmic walking bout detection, utilizing only raw sensor data (see supplementary tables S4). Notably, this approach yielded inferior predictive performance (mobility R² 0.52, ADL R² 0.39, discharge destination accuracy 0.79), suggesting that while imperfect, the walking bout detection algorithm contributes valuable information to the overall model.”

The ultimate goal of the research project is to develop a tool to augment clinical decision-making. Defining acceptable performance thresholds requires careful consideration of the prediction’s risk and the level of human oversight. Fully automated assessments, such as ADL evaluation, would necessitate high accuracy, approaching inter-rater reliability between clinicians (weighted kappa ≥ 0.9). However, even a model with moderate predictive power can be valuable if it aids clinician insights or outperforms current standards of care, as demonstrated by our discharge destination predictions. Future research should focus on refining accuracy, integrating explainability methods for human supervision, and validating performance across diverse clinical applications."

In addition, the supplementary tables were revised to include the additional analysis.

Please expand the discussion of generalizability of models with only N = 39. This is particularly important for the discharge destination model. As mentioned in the limitations, the data set contained only 3 of 4 possible discharge destinations and was heavily skewed (22/39) towards discharge home. Four categories by chance alone is 25%. If the model predicted home for everyone, then accuracy would already be at 56% in this sample.

We agree with the reviewer regarding the limited generalizability of the model given the sample size of n=39 and the class imbalance in discharge destinations. It is important to note, however, that the accuracy of 0.73 observed in standard of care discharge decisions was achieved under the same limitations. While the prevalence of discharge to home introduces a baseline accuracy of approximately 56% by chance alone, accurate prediction of discharge destination is more complex in practice. Differentiating between patients ultimately discharged home versus those requiring higher levels of care necessitates a nuanced understanding of individual risk factors. We have expanded the Limitations section to include a caveat regarding the interpretation of accuracy scores in imbalanced datasets.

The limitation section was expanded as follows:

Third, a significant limitation is the substantial class imbalance observed in discharge destinations, with no patients recommended for nursing home care. This imbalance poses challenges for model training and requires cautious interpretation of accuracy scores, as a majority class prediction can yield deceptively high overall accuracy. While calibration techniques can mitigate some effects, accurate prediction of minority class cases remains a concern”

Somewhere in the Discussion, it is important to mention that there is utility of the actual functional assessments beyond the scores. These assessments often provide key information at the individual item level (e.g. this person needs assistance for toileting but not for stairs, standardized observation of how someone performs the activity to guide rehabilitation treatment) beyond the total score for care team. If the CHARM is only a 1 item scale by observation, it is going to be hard to justify a complex, digital solution that is easier. This information could be included in the Discussion paragraph starting on line 223, where the authors logically debate the pros and cons of the whole endeavor.

We agree with the reviewer’s insightful point regarding the value of detailed functional assessments. We have added a note to the discussion section acknowledging that functional assessments provide detailed clinical information beyond composite scores, which informs individualized care planning:

Furthermore, an inter-rater analysis demonstrated moderate to strong agreement (weighted kappa ≥ 0.80) between predicted and actual assessment scores [23], highlighting the potential for automated functional assessment in geriatric inpatients post-surgery, although it should be noted that functional assessments provide detailed clinical information beyond a composite score, informing individualized care planning”

Minor
In Table 1, this reviewer is highly appreciative of the descriptive statistics from many clinical scales and other aspects of care. There is no information about these scales in the Methods section. Providing more information about the scales would make the paper more accessible to those are not familiar with them or with clinical care. Ideally, the Methods would include a few short sentences about each one. If space does not permit that, then another option would be to add the appropriate references to the abbreviations at the top of Table 1.

We agree with the reviewer that providing additional information about the clinical scales used in this study would enhance the manuscript’s accessibility. We have included a brief description of several key metrics within the revised Methods section. A more thorough description of these assessments can be found in the study protocol for the SURGE-Ahead observational and AI development study [14].

The following addition was made to the methods section:

Baseline characteristics were assessed using several standardized tools. Frailty and multimorbidity were evaluated using the Clinical Frailty Scale (CFS), American Society of Anesthesiologists (ASA) score, Identification of Seniors at Risk (ISAR) score, and number of medications. Cognitive function was assessed with the Montreal Cognitive Assessment 5-min (MoCA) and dementia status was recorded. Nutritional status was assessed using the Nutritional Risk Screening (NRS) score. Psychological distress was assessed with the Patient Health Questionnaire-4 (PHQ-4). Functional status was measured using the Barthel Index, and mobility was further assessed using the Charité Mobility Index (CHARMI) and the New Mobility Score (NMS). Additional data collected included age, sex, type of surgery, emergency surgery status, body mass index (BMI), care level, and history of falls (for details, see study protocol [14]).”

Consider shifting the focus in the text from highlighting statistical significance to highlighting the actual model statistics with their wide confidence intervals. Since a goal is to move forward to clinical care, then the relevance is not with the p value itself but with the variance explained and the range of error at the individual level.

We agree with the reviewer’s suggestion to emphasize model statistics and their associated uncertainty over statistical significance. We have expanded the Limitations section to address this point, including confidence intervals:

First, the relatively small sample size (n=39) limits the generalizability of our findings. Consequently, the reported R² and AUC values should be interpreted with consideration of the wide confidence intervals, which accurately reflect the uncertainty associated with these estimates.”

Reviewer 2 Report

Comments and Suggestions for Authors

The study presents a proof of concept of predicting post-operative mobility scores and recommended discharge destination using a single IMU placed on a patient’s back. The methods appear sound, and the results are interesting. This reviewer has a few comments with the hope of improving the work even further.

  1. “…with substantial to almost perfect agreement with human assessments (weighted kappa ≥ 0.80).” Could the authors clarify if kappa >= 0.80 is considered “almost perfect agreement?” This seems to be an overstatement.
  2. How is the IMU position on the lumbar? Please include a graphic of the sensor relative to the patient, detailing which direction each of the axis of the accelerometer is oriented. Is the y-axis pointing towards the sky? This could help investigate why the y-axis is best in predicting the outcomes specified.
  3. Is the IMU simply an accelerometer? Or does it also include gyroscope and magnetometer?
  4. Please report the part number of the IMU for transparency and reproducibility.
  5. What are the technical and configuration specifications of the IMU? Sample rate? Resolution? Measurement range? Power consumption? How often does it need to be charged?
  6. Figure S2 is very low-res. Can that be improved for readability? And shouldn’t the figure be labeled as a table instead?
  7. Is the recommendation for discharge destination by the two geriatricians considered the gold standard? If so, does the following statement undermine your choice of gold standard: “…the challenges of optimal discharge planning in older adults and the frequent occurrence of undertreatment.”
  8. Why was the lumbar location chosen for the IMU? This seems very inconvenient for patients. “…patient feedback indicated discomfort with lumbar sensor placement, leading several participants (n=7) to decline its use.” Also, are the 7 participants individuals who declined to be in the study at all? Or were they originally in the study but then dropped out?
  9. How long does it take to attach the sensor? This seems to be a very major point that isn’t quite addressed sufficiently. “Furthermore, the time required for accurate lumbar attachment may offset the efficiency gains afforded by automated assessment and discharge decision support.”
  10. What were the predicted recommended discharge destinations for the x and z axes of the IMU?
  11. For the 18% of cases that have missed predictions, can the authors offer some insight as to why the algorithm may have misidentified those cases?
  12. It appears participants wore the IMU for different lengths of time. Does that confound your discharge destination prediction or the various scores? “IMU sensors were worn for an average of 3.10 days (±1.41) following surgery, starting on post-operative day 1.” As in, if you only include participants who wore the IMU for 4 or 5 days, does that improve your algorithm? Or, is it easier to predict discharge destination because the IMU was only worn for 2 days? As in, “if the participant only wore the IMU for two days, that means they were feeling good and were discharged home”, essentially potentially biasing and confounding the algorithm? The algorithm in that case isn’t basing its prediction on IMU data, but just how long the sensor was worn. With that, for the 18% of misidentified cases, how long did they wear the IMU?

Author Response

The study presents a proof of concept of predicting post-operative mobility scores and recommended discharge destination using a single IMU placed on a patient’s back. The methods appear sound, and the results are interesting. This reviewer has a few comments with the hope of improving the work even further.

We appreciate the reviewer’s positive assessment of our study and their constructive comments, which we believe will help to further improve the work

“…with substantial to almost perfect agreement with human assessments (weighted kappa ≥ 0.80).” Could the authors clarify if kappa >= 0.80 is considered “almost perfect agreement?” This seems to be an overstatement.

We thank the reviewer for pointing out a potential overstatement in our interpretation of the kappa statistic. We initially categorized kappa values based on Cohen’s definition as described in McHugh and colleagues (2012), as cited in the manuscript. However, we recognize that in the context of this study, and particularly given the limited sample number, this categorization may be overly optimistic. We have revised the manuscript to reflect a more conservative interpretation, adopting the alternative categorization scheme outlined in McHugh and colleagues (2012) Table 3. The respective sections have been revised as follows:

"The TabPFN model achieved R² values of 0.65 and 0.70 for predicting CHARMI and Barthel Index scores, respectively, with moderate to strong agreement with human assessments (weighted kappa ≥ 0.80)."

"Furthermore, an inter-rater analysis demonstrated moderate to strong agreement (weighted kappa ≥ 0.80) between predicted and actual assessment scores [23], highlighting the potential for automated functional assessment in geriatric inpatients post-surgery, although it should be noted that functional assessments provide detailed clinical in-formation beyond a composite score, informing individualized care planning."

How is the IMU position on the lumbar? Please include a graphic of the sensor relative to the patient, detailing which direction each of the axis of the accelerometer is oriented. Is the y-axis pointing towards the sky? This could help investigate why the y-axis is best in predicting the outcomes specified.

Is the IMU simply an accelerometer? Or does it also include gyroscope and magnetometer?

Please report the part number of the IMU for transparency and reproducibility.
What are the technical and configuration specifications of the IMU? Sample rate? Resolution? Measurement range? Power consumption? How often does it need to be charged?

We thank the reviewer for raising these important points regarding the IMU specifications and placement. We agree that providing more detail will enhance the transparency and reproducibility of our work. Please note a typographic error was identified and corrected regarding the reported association between the IMU axes and outcomes; the manuscript previously stated the z-axis exhibited the highest correlation, when in fact it is the y-axis (see General Revisions). We have added an image to the manuscript illustrating the sensor’s position on the lumbar spine and detailing the orientation of its axes. We have also revised the Methods section to include detailed specifications of the sensor used, as requested:

A subset of 39 participants wore an IMU (Axivity AX6; axivity Ltd., Newcastle upon Tyne, UK) in the lumbar region for at least 24 hours, with 22 concurrently completing a movement diary. Sensors were attached to the participant’s lumbar spine using an adhesive patch on postoperative day 1 with recording starting at 00:00 the following day and detached on postoperative day 5 or earlier if the patient was discharged. The x-axis of the sensor was oriented vertically, the y-axis horizontally, and the z-axis pointed towards the patient’s abdomen, as illustrated in the supplementary figure S2. The sensor operated at a sampling rate of 100 Hz, with a resolution of 16 bits, and a battery lasting approximately 7 days. The Axivity AX6 contains a tri-axial accelerometer and gyroscope; however, in the current study, only the accelerometer data were analyzed.”

Figure S2 is very low-res. Can that be improved for readability? And shouldn’t the figure be labeled as a table instead?

We agree that the former Figure S2 was difficult to read. We have reformatted the TRIPOD+AI Checklist as supplementary table S5 for improved readability.

Is the recommendation for discharge destination by the two geriatricians considered the gold standard? If so, does the following statement undermine your choice of gold standard: “…the challenges of optimal discharge planning in older adults and the frequent occurrence of undertreatment.”

We appreciate the reviewer highlighting this important point regarding the ‘gold standard’ for discharge destinations. It is important to clarify that the standard of care is as follows: a geriatric liaison service is consulted for approximately 15% of patients to provide a recommendation regarding the optimal discharge destination, while the surgeon makes the decision independently for the remaining patients. In the context of the SURGE-Ahead observational and AI development study, we established a more rigorous ‘ground truth’ to enable accurate model training and evaluation. This involved (1) a geriatric assessment for every patient, (2) a follow-up call with the patient, (3) a 3-month chart review, and (4) a consensus decision on the optimal discharge destination reached by two experienced geriatricians. This process surpasses the usual standard of care in several respects, particularly as it incorporates post-discharge outcome data – information that is often unavailable in routine clinical practice. The reviewer’s observation regarding the frequent occurrence of undertreatment reflects the reality that limited resources often prevent comprehensive geriatric co-management for all patients. This is further underscored by the relatively low accuracy of 0.73 observed when comparing our ‘ground truth’ to the standard-of-care discharge decisions. However, while a detailed analysis of discharge destination decisions is beyond the scope of this manuscript, preliminary findings from the SURGE-ahead observational and AI development study are available on GitHub at: https://github.com/IfGF-UUlm/SA_OKIE/ (also see data availability statement), with a full report forthcoming.

Why was the lumbar location chosen for the IMU? This seems very inconvenient for patients. “…patient feedback indicated discomfort with lumbar sensor placement, leading several participants (n=7) to decline its use.” Also, are the 7 participants individuals who declined to be in the study at all? Or were they originally in the study but then dropped out?

We thank the reviewer for raising this important point regarding patient comfort and sensor placement. The decision to utilize a lumbar placement was driven by the requirements of the Mobilise-D algorithm, which was initially developed and validated using data collected from this anatomical location. While we acknowledge the potential for discomfort, initial testing with outpatient populations indicated acceptable levels of comfort. However, as detailed in the manuscript, the prevalence of postoperative immobilization and pain presented a significant challenge to sensor application within our inpatient cohort. We have revised the discussion section to provide further clarification on this issue:

During the observational and AI development study [14], approximately one-quarter (n=46) agreed to wear an IMU sensor, others declined participation due to postoperative immobilization and pain that hindered sensor application. During the sensor application process, several participants (n=7) expressed discomfort with lumbar sensor placement. Importantly, these seven participants were initially enrolled in the study and consented to IMU wear; however, due to discomfort, they elected to wear the IMU sensor on their right thigh instead. Data collected from the thigh-mounted sensors were not included in this analysis, as the Mobilise-D algorithm is currently optimized for data acquired from the lumbar region. Future work will focus on expanding the algorithm’s capabilities to accommodate data from alternative anatomical locations.”

In addition, the supplementary tables were revised to include the additional analysis.

How long does it take to attach the sensor? This seems to be a very major point that isn’t quite addressed sufficiently. “Furthermore, the time required for accurate lumbar attachment may offset the efficiency gains afforded by automated assessment and discharge decision support.”

The entire process – including sensor attachment, data readout, and detachment – typically takes between 10 and 20 minutes, depending on the patient’s mobility and capacity to cooperate. We acknowledge that this time commitment is a potentially significant barrier to implementation, and we have revised the manuscript to reflect this more accurately:

Furthermore, the time required for sensor attachment and data retrieval – typically 10-20 minutes – represents a potential limitation and must be taken into account when evaluating the overall efficiency gains afforded by automated assessment and discharge decision support.”

What were the predicted recommended discharge destinations for the x and z axes of the IMU?

We thank the reviewer for this insightful question. Please note a typographic error was identified and corrected regarding the reported association between the IMU axes and outcomes; the manuscript previously stated the z-axis exhibited the highest correlation, when in fact it is the y-axis (see General Revisions). It is important to clarify that while the Kruskal-Wallis test demonstrated statistically significant differences in IMU data across discharge destination groups, it does not directly establish associations between individual axes and specific destinations. To address the reviewer’s query, we conducted additional analyses, calculating the mean values and standard deviations for the x-, y-, and z-axis stratified by discharge destination. These results are now included in the supplementary table S3. Our findings indicate that the x and y axes did not exhibit consistent or significant associations with recommended discharge destinations, unlike the strong correlations observed with the z-axis.

We have incorporated this information into the discussion section:

In contrast, analyses of the x- and y-axis did not reveal comparable associations with discharge destinations (see supplementary table S3)”

For the 18% of cases that have missed predictions, can the authors offer some insight as to why the algorithm may have misidentified those cases?

We thank the reviewer for raising this important point regarding the 18% of cases with misidentified predictions. Determining the specific reasons for these errors is challenging due to the inherent complexity of deep learning models and the limitations in model explainability, as discussed in the manuscript’s limitations section. In general, overlapping distributions of key variables between different outcome groups can make accurate prediction difficult. However, without a detailed model introspection – which is beyond the scope of this study – attributing specific misclassifications remains largely speculative.

To acknowledge this limitation, we have added the following text to the manuscript:

Finally, explainability of the model’s predictions was not explored and remains an important area for future research. Providing clinicians with a satisfactory understanding of the model’s reasoning as well as possible reasons for inaccurate predictions will be crucial for establishing trust and facilitating clinical adoption.”

It appears participants wore the IMU for different lengths of time. Does that confound your discharge destination prediction or the various scores? “IMU sensors were worn for an average of 3.10 days (±1.41) following surgery, starting on post-operative day 1.” As in, if you only include participants who wore the IMU for 4 or 5 days, does that improve your algorithm? Or, is it easier to predict discharge destination because the IMU was only worn for 2 days? As in, “if the participant only wore the IMU for two days, that means they were feeling good and were discharged home”, essentially potentially biasing and confounding the algorithm? The algorithm in that case isn’t basing its prediction on IMU data, but just how long the sensor was worn. With that, for the 18% of misidentified cases, how long did they wear the IMU?

We thank the reviewer for this excellent and insightful question. The variation in IMU wearing time is indeed a potential confounding factor, and we proactively reported this data in the supplementary table S2. We agree that clinicians would reasonably suspect that wearing time might serve as a proxy for length of hospital stay and thus unfairly influence the prediction model. However, our analysis revealed that IMU wearing time was not significantly associated with discharge destination or the various scores used in our analysis. This suggests that the algorithm’s predictions are driven by the IMU data itself, rather than simply the duration of sensor use, which is a noteworthy result.

To address this point explicitly, we have added the following to the discussion section:

Analysis of IMU wearing time revealed no significant correlation with discharge destination or outcome scores, suggesting that the algorithm’s predictive performance is not primarily driven by the length of sensor use, but rather by the kinematic data acquired during that period (see supplementary table S2).”

Reviewer 3 Report

Comments and Suggestions for Authors

The paper presents a deep learning based post operative mobility prediction techniques using IMU data. An actual clinical trials witht the involvement of elderly patients recovering form surgeries were used. They have used manual entries as reference. They have reported higher detection accuracies.

However, there are few concerns that needs to be addressed.

manual event recording by the patient is not very accurate. This affects the accuracy of the results. It is better to use another autonomous techniques to verify the events, maybe image based.

The accuracy of the IMU data must be elaborated. There should be either accuracy data from previous studies, datasheets or if not, they must present accuracy data.

Author Response

The paper presents a deep learning based post operative mobility prediction techniques using IMU data. An actual clinical trials with the involvement of elderly patients recovering from surgeries were used. They have used manual entries as reference. They have reported higher detection accuracies.

However, there are few concerns that needs to be addressed.

manual event recording by the patient is not very accurate. This affects the accuracy of the results. It is better to use another autonomous techniques to verify the events, maybe image based.

We thank the reviewer for this insightful suggestion regarding the potential for improved event verification. We agree that manual patient event recording is inherently susceptible to inaccuracies and that autonomous techniques, such as image-based analysis, could offer a more objective approach. However, we must clarify that the collection of such data was not feasible within the scope of the SURGE-Ahead observational and AI development study due to ethical considerations and concerns regarding patient data privacy. We acknowledge that this reliance on patient-reported events likely introduces a degree of bias and may contribute to reduced accuracy in our results. We have incorporated this limitation into the manuscript’s limitations section, as follows:

Fourth, the reliance on patient-reported events, which are subject to bias and potential inaccuracies, is a potentially limiting factor; the absence of objective, autonomous event verification techniques, such as image-based analysis, may have impacted the accuracy of the results.”

The accuracy of the IMU data must be elaborated. There should be either accuracy data from previous studies, datasheets or if not, they must present accuracy data.

We thank the reviewer for highlighting the importance of detailing the accuracy of the IMU data. We agree that providing this information enhances the transparency and reproducibility of our work. We have expanded the Methods section to include further details regarding the sensor’s specifications:

A subset of 39 participants wore an IMU (Axivity AX6; axivity Ltd., Newcastle upon Tyne, UK) in the lumbar region for at least 24 hours, with 22 concurrently completing a movement diary. Sensors were attached to the participant’s lumbar spine using an adhesive patch on postoperative day 1 with recording starting at 00:00 the following day and detached on postoperative day 5 or earlier if the patient was discharged. The x-axis of the sensor was oriented vertically, the y-axis horizontally, and the z-axis pointed towards the patient’s abdomen, as illustrated in the supplementary figure S2. The sensor operated at a sampling rate of 100 Hz, with a resolution of 16 bits, and a battery lasting approximately 7 days. The Axivity AX6 contains a tri-axial accelerometer and gyroscope; however, in the current study, only the accelerometer data were analyzed.”

Regarding the accuracy of walking bout detection, we have included a comparison to values reported in the literature, specifically referencing the work of Kirk and colleagues (2024). We acknowledge that the prediction of discharge destination using sensor data is a relatively novel application of AI and machine learning. While we typically refrain from explicitly emphasizing novelty, we believe it is important to contextualize our findings. Therefore, we have added the following sentence to the Discussion section:

A single lumbar IMU sensor and deep learning also accurately predicted the recommended discharge destination in 82% of cases, representing a novel application of IMU data analysis.”

Furthermore, we report standard-of-care metrics within the Discussion section, and while a detailed analysis of discharge destination decisions is beyond the scope of this manuscript, preliminary findings from the SURGE-ahead observational and AI development study are available on GitHub at: https://github.com/IfGF-UUlm/SA_OKIE/ (also see data availability statement), with a full report forthcoming.

Reviewer 4 Report

Comments and Suggestions for Authors

The idea and results of this study are quite clear. However, several questions arose when reading the text:

  1. Why was the IMU sensor mounted in the lumbar region? It is located quite low to the ground and therefore produce smaller spatial shift. Given the "pendulum" model of human posture and gait, the best place to detect movement is the extreme point of the pendulum, namely, the head when walking (in z-axis). Also, z-axis looks convenient to detect vertical movements (standing up). On the other hand, the head can move in a lying or sitting position, without walking, which can make it difficult to detect real locomotion episodes. In addition, the lumbar position is more comfortable for wearing sensors. So I would agree that the lumbar position was relevant.
  2. Was a gyroscope used along with the IMU accelerometer? Usually, IMUs are equipped with both. A gyroscope would be useful for detecting rotational accelerations (turns, etc.), which are part of real walking in (e.g. bypassing furniture, walking on the street as a pedestrian, etc.).
  3. Line 95. Feature extraction section: were only walking episodes detected and analyzed? Were the number of steps and locomotion characteristics (e.g. step length, stepping cadence, walking speed, etc.) taken into account? Also, variability of stepping characteristics would be important. For example, in older people stepping is more variable in the frontal (coronal) plane (medial-lateral movement).

Author Response

The idea and results of this study are quite clear. However, several questions arose when reading the text:

1. Why was the IMU sensor mounted in the lumbar region? It is located quite low to the ground and therefore produce smaller spatial shift. Given the "pendulum" model of human posture and gait, the best place to detect movement is the extreme point of the pendulum, namely, the head when walking (in z-axis). Also, z-axis looks convenient to detect vertical movements (standing up). On the other hand, the head can move in a lying or sitting position, without walking, which can make it difficult to detect real locomotion episodes. In addition, the lumbar position is more comfortable for wearing sensors. So I would agree that the lumbar position was relevant.

We thank the reviewer for this insightful discussion regarding sensor placement and its impact on data acquisition. We concur that the lumbar region presents a compromise between capturing meaningful movement and ensuring patient comfort. The selection of the lumbar region was initially dictated by the requirements of the Mobilise-D algorithm [Micó-Amigo et al., 2024], which served as the foundation for our analysis. However, we acknowledge the reviewer’s point regarding the potential advantages of alternative locations, particularly the head, for detecting gait parameters and vertical movements. Interestingly, recent work by Kirk and colleagues (2024) has demonstrated meaningful detection of functional decline using lumbar-worn IMU data, supporting the validity of our chosen approach. To further investigate the contribution of walking bout detection to our model’s performance, we conducted an additional analysis excluding algorithmic data and relying solely on raw sensor readings. As detailed in the revised manuscript, this approach resulted in inferior predictive performance (mobility R² 0.52, ADL R² 0.39, discharge destination accuracy 0.79). These findings suggest that, despite its limitations, the walking bout detection algorithm adds valuable information to the overall predictive model. The manuscript has been revised to reflect this analysis, and the results are presented in supplementary table S4.

We also performed an additional analysis excluding algorithmic walking bout detection, utilizing only raw sensor data (see supplementary table S4). Notably, this approach yielded inferior predictive performance (mobility R² 0.52, ADL R² 0.39, discharge destination accuracy 0.79), suggesting that while imperfect, the walking bout detection algorithm contributes valuable information to the overall model.”

2. Was a gyroscope used along with the IMU accelerometer? Usually, IMUs are equipped with both. A gyroscope would be useful for detecting rotational accelerations (turns, etc.), which are part of real walking in (e.g. bypassing furniture, walking on the street as a pedestrian, etc.).

We thank the reviewer for raising this point. As now detailed in the revised Methods section, the Axivity AX6 contains both a tri-axial accelerometer and gyroscope; however, in this study, only accelerometer data were analyzed, because the Mobilise-D algorithm does not use gyroscope data.

A subset of 39 participants wore an IMU (Axivity AX6; axivity Ltd., Newcastle upon Tyne, UK) in the lumbar region for at least 24 hours, with 22 concurrently completing a movement diary. Sensors were attached to the participant’s lumbar spine using an adhesive patch on postoperative day 1 with recording starting at 00:00 the following day and detached on postoperative day 5 or earlier if the patient was discharged. The x-axis of the sensor was oriented vertically, the y-axis horizontally, and the z-axis pointed towards the patient’s abdomen, as illustrated in the supplementary figure S2. The sensor operated at a sampling rate of 100 Hz, with a resolution of 16 bits, and a battery lasting approximately 7 days. The Axivity AX6 contains a tri-axial accelerometer and gyroscope; however, in the current study, only the accelerometer data were analyzed.”

3. Line 95. Feature extraction section: were only walking episodes detected and analyzed? Were the number of steps and locomotion characteristics (e.g. step length, stepping cadence, walking speed, etc.) taken into account? Also, variability of stepping characteristics would be important. For example, in older people stepping is more variable in the frontal (coronal) plane (medial-lateral movement).

We thank the reviewer for this important question regarding the feature extraction process. As detailed in the Methods section, the Mobilise-D algorithm not only detects walking episodes but also calculates a comprehensive set of locomotion characteristics, including cadence, stride duration, and walking speed, as well as their associated variance and maximum values. To provide a complete overview of the variables derived from the Mobilise-D pipeline, we have added a statement to the manuscript indicating that all 94 extracted features are reported in the supplementary tables.

The Mobilise-D pipeline identifies walking bouts and calculates their duration, cadence, stride duration, and walking speed, along with their average, variance, and maximum values (see supplementary table S2). For more information on the Mobilise-D project, please refer to the project page (https://mobilise-d.eu/) and associated publications [8,9].”

Round 2

Reviewer 1 Report

Comments and Suggestions for Authors

Thank you for the thoughtful and thorough revisions.